# Profiling of Phenolic Compounds and Triterpene Acids of Twelve Apple (*Malus domestica* Borkh.) Cultivars

**DOI:** 10.3390/foods10020267

**Published:** 2021-01-28

**Authors:** Elisabeta-Irina Geană, Corina Teodora Ciucure, Roxana Elena Ionete, Alexandru Ciocârlan, Aculina Aricu, Anton Ficai, Ecaterina Andronescu

**Affiliations:** 1National Research and Development Institute for Cryogenics and Isotopic Technologies—ICSI Rm. Valcea, 4th Uzinei Street, PO Raureni, Box 7, 240050 Rm Valcea, Romania; irina.geana@icsi.ro (E.-I.G.); corina.ciucure@icsi.ro (C.T.C.); roxana.ionete@icsi.ro (R.E.I.); 2Faculty of Applied Chemistry and Materials Science, Politehnica University of Bucharest, 1-7 Polizu Str., 011061 Bucharest, Romania; anton.ficai@upb.ro; 3Institute of Chemistry, Academiei Str. 3, MD-2028 Chisinau, Moldova; algciocarlan@yahoo.com (A.C.); aculina.aricu@gmail.com (A.A.); 4Academy of Romanian Scientists, Ilfov St. 3, 050045 Bucharest, Romania

**Keywords:** bioactive apple compounds, phenolic acids, flavonoids, triterpene acids, UHPLC (ultra high performance liquid chromatography), chemometric analysis

## Abstract

Apple (*Malus domestica* Borkh.), a popular and widely cultivated fruit world-wide, contains bioactive compounds responsible for their health benefits. Here we report the amounts of some bioactive compounds: two major triterpenes (oleanolic and ursolic acids) and polyphenols (phenolic acids, flavan-3-ols, flavonoids and t-resveratrol), together with bioactive properties of twelve apple cultivars measured by chromatographic and spectrophotometric methods. Significant variations were found comparing the bioactive potential of the investigated cultivars. High contents of phenolic acids were identified in the Montuan, Golden Delicious and Cretesc cultivars, while the most flavonoid dominant was the Richard cultivar. Starkrimson, Jonatan, Beliy Naliv and Richard cultivars present higher antioxidant capacity. Oleanolic acid ranged from 11 to 83 mg/g apple extract, while ursolic acid ranged from 55 to 436 mg/g apple extract, with higher amounts in Richard and Montuan cultivars. Principal component analysis (PCA) and hierarchical cluster analysis (HCA) allowed the discrimination of apple cultivars depending on polyphenolic and triterpene acids composition. Caffeic acid, gallic acid and epicatechin were identified as the main bioactive compounds in Starkrimson, Jonathan, Beliy Naliv and Richard cultivars, while ursolic and oleanolic acids were identified in high amounts in Richard, Montuan, Golden Delicious, Idared and Beliy Naliv apple cultivars. The results obtained in this study will contribute to the understanding of the bioactive composition of apples as well as the importance of their capitalization to obtain value-added products that promote human health.

## 1. Introduction

Originating somewhere between the Caspian and the Black Seas, the apple (*Malus domestica* Borkh.) is one of the most extensively produced and consumed fruits worldwide [1], being available on the market for the whole year and, representing the main source of polyphenols in human diet [2,3]. The Health Program in European countries indicates that the nutrition of school-age children should be supplemented with vegetal products rich in vitamins, minerals and numerous bioactive compounds, apples being preferred in this regard [4].

Consumer awareness of the relationship between food and health has led to an increased demand for the consumption of foods rich in antioxidants (like polyphenols) and triterpene acids because of their preventative effect against many diseases. Many studies have associated apples and their nutrients with a positive influence on human health [5]. Biologically active compounds (phenolic compounds and triterpene acids) in apples have shown anti-inflammatory properties [6,7] and protective effect against neurodegeneration and Alzheimer’s disease [8], as well as reducing the risk of cardiovascular disease, lung and colorectal cancers and type II diabetes [8,9].

It is known that phenolic compounds occur in plants in the form of glycosides or esterified with carboxylic acids, and apples have the highest number of free phenolics, among other edible fruits [8]. Phenolic compounds which occur in apples can be divided into several groups: hydroxybenzoic and hydroxycinnamic acids, flavonols, dihydrochalcones, and anthocyanins [10]. Flavonols and anthocyanins are usually found in the peel, while flavanols, dihydrochalcones and hydroxycinnamic acids are the major polyphenol groups found in the apple flesh [11].

The concentration of individual phenolic compounds and triterpene acids in apples is influenced by variety, maturity of the fruit, time of harvest, type of farming, storage, infection development, weather conditions and pedoclimatic conditions [8,12,13]. Moreover, the content of phenolic compounds in apples depends on the part of the apple, particularly, the peel or the flesh. Generally, the content of bioactive phytochemicals is particularly high in the peel compared with the flesh (the peel contains from two to six times more phenolics than the flesh), and therefore the consumption of apples with peel is highly recommended [3,14]. Consequently, apple peel is richer in total phenolic compounds, total flavonoids and total procyanidins than flesh [15] and apples more abundant in phenolic compounds tend to have a higher antioxidant activity [3].

Ursolic and oleanolic acids and other pentacyclic triterpenoids are present in all aerial parts of plants and play an important role in their protection. These compounds may be found in abundance in fruit peel’s intra-cuticular layer localized under the epicuticular waxes [16,17].

Even if the conventional extraction methods, including maceration, percolation and Soxhlet extraction, are still used for the extraction of natural products from different vegetable matrices, some modern and greener extraction methods such as ultrasound extraction, supercritical fluid extraction, pressurized liquid extraction and microwave assisted extraction, have also been applied in this direction [18]. In order to isolate a particular group of compounds, the selection of the extraction solvent according to the solubility of the compounds and the optimization of extraction conditions represent essential steps for developing an efficient extraction process [19].

A significant amount of apple crop is industrially processed, mainly to produce juices, resulting in a large volume of solid residues (consisting of peels, seeds, and pulp), known as “apple pomace” [20]. This by-product of the apple industry represents a valuable source of bioactive compounds and nutrients to be used for the development of new products with nutraceutical properties. Therefore, it is important to select the proper apple cultivars in terms of specific bioactive compounds, so that the resulting pomace has high potential for capitalization.

Considering that apples are consumed as a whole, including both peel and flesh, the aims of this study were to: (i) evaluate and quantify the phenolic compounds (phenolic acids, flavonoids and t-resveratrol) and associated bioactive properties (total polyphenols, total flavonoids and antioxidant capacity), together with the triterpene acid content of different apple cultivars from Romania and the Republic of Moldova in order to obtain a comprehensive view of the content of these bioactive compounds for each cultivar; (ii) discriminate different apple cultivars based on polyphenolic and triterpene acid composition using Principal Component Analysis (PCA) and Hierarchical Cluster Analysis (HCA) and (iii) review the biological and pharmacological activities of the main bioactive compounds identified in the studied apple cultivars, highlighting the potential of different apple cultivars to protect human health.

## 2. Materials and Methods

### 2.1. Apple Samples

In total, fourteen apple samples corresponding to twelve apple cultivars were collected during 2015 harvest season from Romania and the Republic of Moldova, more exactly: Starkrimson (STK), Idared (ID), Cretesc (CR), Golden Delicious (GD RO), Jonathan (IO RO) as cultivars from Romania; and Florina (FL), Richard (RCH), Renet Simirenco (RS), Spartan (SP), Beliy Naliv (BN), Montuan (MO), Gloster (GL), Golden Delicious (GD MD), Jonathan (IO MD) as cultivars from Republic of Moldova. The sample locations were: the Valcea (44°99′ N, 23°82′ E) and Arges (44°79′ N, 24°67′ E) counties from Romania; and Truseni village-Chisinau municipality (47°04′ N, 28°69’ E), Briceni (48°27′ N, 26°94′ E), Edinet (48°16′ N, 27°27′ E), Nisporeni (47°08′ N, 28°16′ E), Ungheni (47°25′ N, 27°89′ E), and Orhei (47°40′ N, 28°76′ E) districts from the Republic of Moldova. Approximately, 2 kg of representative apples were collected for each variety during the October campaign. The sample quantity was divided in two, half of the amount for phenolic determination and the other half for triterpene acids determination. For phenolic compounds determination, the apples were cut into slices of equal size (up to 1 cm in thickness) and the tails and the seeds were removed, followed by the dehydration of apple slices at 40 °C using a Biovita drier (Globus Transport, Cluj Napoca, Romania) and then ground to fine powder by using a Retsch 200 mill (Verder Scientific, Haan, Germany). For triterpene acid determination, whole apples were used without a previous pre-treatment, apart from cleaning their surface.

### 2.2. Chemicals

All the used reagents and solvents were of analytical or liquid chromatography purity and were obtained from Merck (Darmstadt, Germany). Formic and phosphoric acids, anhydrous sodium carbonate, aluminium chloride, sodium acetate and 96% ethanol were analytical grade, while methanol and acetonitrile were for liquid chromatography use. The Folin–Ciocalteu phenol reagent (2 N), 2,2-Diphenyl-1-picrylhydrazyl and 6-hydroxy- 2,5,7,8-tetramethyl-2-carboxylic acid (Trolox) used for the determination of total polyphenols composition and antioxidant activity were purchased from Sigma-Aldrich (St. Louis, MO, USA). All phenolic standards (caffeic, gallic, ferulic, *p*-coumaric, p-hydroxybenzoic, 3,4-dihydroxybenzoic, *t*-cinnamic and chlorogenic acids, catechin, epicatechin, quercetin, rutin, *t*-resveratrol, ursolic and oleanolic acids) had purities corresponding to high performance liquid chromatography (HPLC) and were purchased from Sigma-Aldrich (Steinheim, Germany). Stock and working standard solutions were prepared in methanol. Deionised water, produced by a Milli-Q Millipore system (Bedford, MA, USA), was used for the preparation of aqueous solutions and mobile phases.

### 2.3. Extraction Procedures

Polyphenol extraction was performed using an ultrasonic bath (Elma, Singen, Germany), following the protocol of Jakobek et al. with slight changes [11]. An amount of 0.5 g of apple powder was weighed and 10 mL of 80% acidified methanol was added, followed by ultrasonic extraction for 30 min at 40 °C, ensuring a constant temperature of the water bath by partially replacing the water in the bath with cold water. The obtained extracts were centrifuged, followed by the quantitative recovery of the analytes, filtration and analytical determinations. The selective extraction of triterpene ursolic and oleanolic acids from the cuticular layer of the apples was performed according to a previously described method [21]. Aliquots of each crude extract were dissolved in methanol using ultrasonication and filtered through 0.45 μm PTFE (polytetrafluoroethylene) membrane syringe filters (Merck Millipore, Darmstadt, Germany) before the analysis.

### 2.4. Analytical Investigations

#### 2.4.1. Chromatographic Determinations

Individual polyphenols in extracts were determined by using reverse phase ultra-high performance liquid chromatography with diode array detection (RP-UHPLC-DAD) using a Dionex Ultimate UHPLC 3000 system (Thermo Fisher Scientific Inc., San Jose, CA, USA). Compounds separation was performed on Accuacore PFP (penta-fluorophenyl) column (100 mm × 2.1 mm, 2.6 μm) operated at a constant temperature of 30 °C in the column oven. The mobile phase was a binary solvent system consisting of solvent A (water with 0.1% formic acid) and solvent B (acetonitrile with 0.1% formic acid), both eluted at 0.3 mL/min, after a gradient programme: 0 min, 2% B, 0–9.1 min, 2–50% B; 9.1–9.2 min, 50–60% B; 9.2–9.5 min, 60–65% B; 9.5–10 min, 65–2% B; 10–15 min, 2% B for final washing and equilibration of the column for the next run. Calibration curves of the standards covered the range of 1–50 mg/L for each phenolic compound.

The identification of the chromatographic peaks was achieved by comparing the retention times and spectral characteristics with those of authentic standards. Furthermore, extracts were spiked with polyphenol standards which provide additional information on polyphenol identification. Calibration curves of the standards were made by preparing stock standard solution (100 mg/L in 100% methanol) and working solutions covering the range 1–50 mg/L for each phenolic compound. All stock and working solutions were stored in the dark at 4 °C and were stable for at least three months. Recovery studies were performed by spiking the apple powder with 500 µL of 100 mg/L solution containing each standard phenolic compound, followed by the extraction protocol and instrumental analysis.

Ursolic and oleanolic acids were quantified by a high-performance liquid chromatography-photodiode array (HPLC-PDA) method previously reported [21], using a Thermo Finnigan Surveyor Plus HPLC System (San Jose, CA, USA).

#### 2.4.2. Quantitative UV-Vis Spectrophotometric Determinations

Colorimetric spectrophotometric determinations of the apple extracts (total polyphenols—TP, total flavonoids—TF and antioxidant capacity—AC) were performed using a Specord 250 Plus UV-Vis spectrophotometer (Analytic Jena, Germany) equipped with 1 cm path length quartz cells.

Total polyphenols (TP) were determined by the Folin-Ciocalteau method [22], measuring the maximum absorbance at 675 nm. In brief, 100 μL of the apple extract was added to test tubes and mixed with 5 mL ultrapure water and 200 µL Folin Ciocalteau reagent. After 5 min of reaction, 300 µL of 20% sodium carbonate solution was added to stop the reaction and to develop a characteristic blue colour for 2 h, at room temperature and protected from light. The total phenolic quantification was based on the standard curve generated by serial dilution of a gallic acid standard, covering the range 50–1000 mg/L of gallic acid. Values were expressed as mg gallic acid equivalents (GAE) per 100 g of apple, dried weight (DW).

Total flavonoids (TF) were determined by the aluminium chloride method described by Hosu et al. (2016) [23], with some adaptations. 0.5 mL of apple extract were treated with 0.4 mL of 25 g/L AlCl3 solution, 0.5 mL of 100 g/L sodium acetate solution and 4 mL distilled water. After 15 min, the maximum absorbance was measured at 430 nm. The total flavonoids content was expressed as mg rutin equivalent (RE) per 100 g of apple (DW) based on the calibration curve obtained for rutin in the 0–125 µg/mL concentration range.

Antioxidant capacity (AC)—DPPH method: synthetic radical, 2,2-diphenyl-1-picrylhydrazyl (DPPH) was used to determine the AC of the apple extracts. In this assay, 6 mL of 0.09 mg/mL DPPH methanolic solution was mixed with 0.5 mL aliquots of apple extract and the absorbance was measured at 517 nm, after 20 min at room temperature. A blank control of ethanol/water mixture was run for each assay. Absorbance measurements were transformed into antioxidant capacity using trolox as standard, in the concentration range 50–1000 µmol/L and the results were expressed as µmol/L Trolox equivalents/100 g of apples DW.

### 2.5. Statistical Data Processing Methods

All the experiments were carried out in duplicate and the obtained values were expressed as means and standard deviations. The obtained analytical data were processed statistically by analysis of variance (ANOVA) and used to evaluate significant differences among different apple cultivars. The Duncan’s test was used to discriminate the different apple varieties (differences at *p* ≤ 0.05 were considered to be significant). Principal Component Analysis (PCA) and Hierarchical Clustering Analysis (HCA) were also performed in order to discriminate between different apple varieties. All the mathematical and statistical analyses were performed using Microsoft Excel 2010 and XLSTAT Add in soft version 15.5.03.3707.

## 3. Results and Discussions

### 3.1. Polyphenolic Composition

The performances of the analytical protocol used for UHPLC quantification of phenolic compounds are detailed in Appendix A. Calibration curves revealed good linearity, with *R*^2^ coefficients higher than 0.995. Limits of detection (LODs) ranged from 0.09–0.41 mg/100 g, whereas limits of quantification (LOQs) ranged from 0.20–1.32 mg/100 g. For each quantified phenolic compound, the recovery rates were between 65–75% for phenolic acids, 67–85% for flavonoids and 94% for *t*-resveratrol, while the precision values were <5%.

The content of individual phenolic acids, flavonoids and *t*-resveratrol of the investigated apple varieties are presented in Table 1 and Table 2. The obtained data are expressed as mean values and standard deviations. With respect to the polyphenolic composition, the results generated showed that the main significant changes were observed for caffeic, gallic and *p*-coumaric acids, epicatechin, rutin and quercetin.

Caffeic, gallic, *p*-coumaric and ferulic acids were the dominant acids among the investigated phenolic acids, with values ranged from 9.43–47.23 mg/100 g DW for caffeic acid, 0.49–21.74 mg/100 g DW for gallic acid, 1.09–16.31 mg/100 g DW for *p*-coumaric acid and 0.39–7.27 mg/100 g DW for ferulic acid. Even if chlorogenic acid in apples was reported in high amounts in different studies [3,8,24], the amount of chlorogenic acid in the investigated apple cultivars varied from not detected to 3.77 mg/100 g DW, similar to the findings of [25].

The literature reports considerably greater amounts of (−)-epicatechin compared with (+)-catechin [26] and this agrees with the results of our study. The ratio of (−)-epicatechin to (+)-catechin in the extracts of different cultivar apple ranged from 2 in the case of Spartan cultivar to 13 in the case of Golden Delicious from the Republic of Moldova.

The data in Table 2 show that the amount of the identified monomeric flavan-3-ols varied from 0.64 to 4.08 mg/100 g DW for (+)-catechin and from LOQ to 25.03 mg/100 g DW for (-)-epicatechin. High amount of catechin was observed for Starkrimson cultivar, while epicatechin was quantified in high amounts in Starkrimson, Florina and Richard cultivars (concentrations about five times higher than the other cultivars).

The amounts of (-)-epicatechin determined in 14 ancient apple cultivars from Friuli Venezia Giulia (Italy) and six commercial apple cultivars varied from 9.10 to 85.39 mg/100 g of peel (FW) and from 7.12 to 27.06 mg/100 g of pulp (FW), while (+)-catechin varied from 3.53 to 20.20 mg/100 g in peel (FW) and from 4.21 to 16.51 mg/100 g of pulp (FW) [27]. Bondonno et al. reported 10.93 mg (-)-epicatechin/100 g DW (whole apple) for the Golden Delicious apple cultivar harvested in Western Australian in 2015 season [28], while Preti et al. reported for Golden Delicious the following values: 31.00 mg (-)-epicatechin/100 g and 10.27 mg (+)-catechin/100 g of apple peel (FW) and respectively, 7.49 mg (-)-epicatechin/100 g and 4.21 mg/100 g of apple pulp (FW) [27].

The concentration of quercetin found in the apples ranged between 1.10–11.61 mg/100 g DW with higher amounts for Beliy Naliv and Spartan cultivars and lowest amounts for Renet Simerenco and Richard cultivars. Rutin was a minor component among the other flavonoids, with highest concentration for Beliy Naliv, Montuan and Cretesc cultivars, being consistent with the literature [27,29].

*t*-Resveratrol was quantified in a high amount in Beliy Naliv, Starkrimson and Montuan cultivars, and the lowest quantities were detected in Idared and Renet Simirenco cultivars. The content of individual polyphenols (Table 1 and Table 2) is similar to other literature data [3,29,30].

The concentration of different groups of phenolic compounds varied widely from cultivar to cultivar (Figure 1). According to the results, the studied apple cultivars are phenolic acid dominated, except the Richard cultivar which is flavonoids dominated. The cultivars with the highest level of phenolic acids were ‘Montuan’ (55.03 mg/100 g DW), Golden Delicious (MD) (47.53 mg/100 g DW), Cretesc (45.69 mg/100 g DW), and Spartan (44.98 mg/100 g), while the cultivar with the lowest level of phenolic acids was Richard (17.55 mg/100 g). A high quantity of total flavonoids was observed for the Starkrimson cultivar (33.27 mg/100 g), while a smaller amount was observed for the Cretesc cultivar (6.83 mg/100 DW). The Florina cultivar shows similar amounts of phenolic acids (25.12 mg/100 g DW) and flavonoids (24.65 mg/100 g DW).

The values of the TP obtained by the Folin Ciocalteu colorimetric method ranged from 240–576 mg GAE/100 g DW, while TF ranged between 65–166 mg RE/100 g DW (Appendix A). Starkrimson, Richard and Idared were the apple cultivars with a high TP content, while Montuan, Jonathan and Spartan were the apple cultivars with high TF content. TP content of different apple cultivars harvested in Italy varied from 156 to 535 mg GAE/100 g peel (DW) and from 46 to 175 mg GAE/100 g pulp (DW) [27], while TF content of different apple cultivars harvested in Korea varied from 104 to 192 mg QE/100 g of peel (DW) and from 1.5 to 3.2 mg QE/100 g of pulp (DW), respectively [31].

The results of the AC of all the investigated apple cultivars measured by DPPH assay demonstrated a large variability, ranging from 1277 to 2794 µmol Trolox/100 g DW. Extracts from Starkrimson, Jonathan, Richard and Beliy Naliv cultivars showed the highest AC, whereas the Gloster, Montuan, Golden Delicious and Spartan revealed the lowest AC. The values of AC of the peel and pulp (DW) of different apple cultivars harvested in Korea, measured by DPPH assay, varied between 280–500 µmol Trolox equivalents/100 g of peel (DW) and between 12–348 µmol Trolox equivalents/100 g of pulp (DW) [32]. Thus, the cultivars that can be highlighted by their high content of polyphenols, being a good source of polyphenols, are Starkrimson, Jonathan, Richard and Beliy Naliv.

The results of TP and TF obtained by colorimetric methods were higher compared with UHPLC quantification results, being in concordance with the findings of [30], indicating that there are more phenolic compounds, including phloretin, phloridzin, polymeric procyanidins (procyanidin B1, procyanidin B2), quercetin-glycosides [33,34] and others, which contributed to the total polyphenolic content and which were not quantified in this study because we did not have the necessary standards.

The ANOVA results applied to the spectrophotometric results indicate that the TP, total TF and AC were significantly different depending on the apple cultivars (Appendix A). Higher TP content was identified for Starkrimson, Richard and Idared cultivars, and higher TF content for Beliy Naliv and Montuan cultivars.

The correlation analysis (Table 3) shows low to moderate correlations between the UHPLC phenolic compounds profile (expressed as ∑ phenolic acids and ∑ flavonoids) and bioactive properties of the investigated apple cultivars (TP, TF and AC). The interpretation of correlation analysis was done using correlation coefficients with values higher than 0.5.

Positive low correlations were obtained for TP with phenolic acids’ UHPLC profile (∑ phenolic acids), AC with flavonoids’ UHPLC profile (∑ flavonoids) and AC with TF, while moderate positive correlation was observed for TF with flavonoids’ UHPLC profile (∑ flavonoids). These low and moderate correlations can be explained by the fact that, beside the phenolic compounds determined in this study by UHPLC analysis, in apples there are also other phenolic compounds which contribute to TP, TF and AC.

### 3.2. Triterpene Acids Composition

The content of oleanolic and ursolic acids in the extracts obtained from the cuticular layer of different apple cultivars is presented in Figure 2.

In apple extracts, the ursolic acid was quantified in higher amounts compared with oleanolic acid, being consistent with the literature data [6,33]. The content of oleanolic acid in the extracts obtained from apple cuticular layer varied from 10.88–82.53 mg/g extract, with higher amounts for Richard, Montuan, Golden Delicious (MD) and Idared cultivars and lower content for Starkrimson, Cretesc and Jonathan cultivars. The ursolic acid content ranged from 54.68 to 435.57 mg/g extract, with higher content for Richard, Montuan, Idared and Beliy Naliv cultivars and lower content for Starkrimson and Gloster cultivars. The ANOVA results indicate that the content of ursolic and oleanolic acids in apple cuticular layer were significantly different depending on the apple cultivars.

### 3.3. Apple Cultivar Discrimination Based on Polyphenolic and Triterpene Contents

For exploratory data analysis purposes, PCA was performed on the polyphenolic quantitative UHPLC and UV-Vis data. The first two principal components (PC1 and PC2) with 46.71% of the whole variances were extracted for analysis. PC1 accounted for 26.39% variances and PC2 accounted for 20.32%. The distribution of the apple cultivars in the PC1-PC2 score plot is presented in Figure 3.

It can be observed that there is no clear discrimination between the apple cultivars, but several trends were observed. Thus, three distinct clusters distributed in the four quadrants of PC1-PC2 space were noticed. Similar apple cultivars appeared located at the same part of the PC1 axis, Richard, Florina, Idared, Renet Simirenko, Jonathan and Golden Delicious from Romania, Montuan and Gloster cultivars being located on the right side, and Starkrimson, Beliy Naliv, Cretesc, Spartan, Jonathan and Golden Delicious from Republic of Moldova at the left side of the PC1 axis. Our results showed that t-resveratrol, chlorogenic and 3,4-dyhydroxibenzoic acids, catechin together with TP and AC could be suggested as polyphenolic markers of the Starkrimson cultivar, while ferulic acid and epicatechin represent polyphenolic markers of the Richard, Florina and Idared cultivars. Rutin, quercetin, gallic and *p*-coumaric acids characterise the Beliy Naliv, Spartan and Cretesc cultivars, while caffeic, t-cinnamic and 4-hydroxibenzoic acids are representative for Renet Simirenco and Montuan apple cultivars.

Hierarchical Cluster Analysis was used as an exploratory tool to assess heterogeneity among different apple cultivars based on triterpene composition (Figure 4). At a dissimilarity level of 38,000, class 1 (C1) refers to the Gloster, Starkrimson, Jonathan, Golden Delicious and Cretesc cultivars, while class 2 include the Spartan, Florina and Renet Simirenco cultivars and class 3 (C3) grouped the Beliy Naliv, Idared, Montuan, Richard and Golden Delicious from Republic of Moldova, being the cultivars with higher ursolic and oleanolic acid contents.

The results indicate that cultivars with a higher potential of polyphenols such as Starkrimson, Jonathan and Cretesti represent cultivars with a lower potential of triterpene acids. The cultivars with lower polyphenolic potential such as Montuan and Golden Delicious represent the cultivars with high triterpene acid content. Beliy Naliv, Golden Delicious and Richard apple cultivars shows both polyphenolic and triterpene acid potential.

There are various scientific reports on the biological and pharmacological activities of bioactive compounds from various natural sources with a positive impact on human health, including caffeic and gallic acids, epicatechin, quercetin and ursolic and oleanolic acids, which were identified as the main bioactive compounds in the studied apple cultivars. Table 4 shows the main bioactive compounds identified in the studied apple cultivars and the associated biological and pharmacological activities which contribute to human health, as reported in the scientific literature. Thus, the main biological and pharmacological activities relate to the antioxidant, antimicrobial, anti-inflammatory, anticancer, cardioprotective, gastroprotective, and neuro-protective effects [35,36,37]. Caffeic acid and epicatechin reduce fat accumulation and increase muscle mass, being marketed as a supplement to aid in weight loss and enhance athletic performances, and also contribute to a significant decrease of blood pressure and sugar levels, being used for cardiovascular and diabetic protection [38,39,40]. It should be mentioned that there are not enough clinical studies demonstrating the benefits of ursolic and oleanolic acids to human health and further studies should be concentrated in this direction.

While there are many health benefits to consuming apples in the daily diet as a rich source of bioactive compounds, the contribution made in the protection of human health by daily apple consumption is relatively low. Therefore, daily intake of bioactive compounds into the human body can be completed by using different natural supplements and functional foods based on bioactive extracts from natural sources, including apples and the by-products resulting from apple processing.

The presented results can be useful for selective industrial apple processing, so that the by-products resulting from apple juice production serve as natural source for the isolation of the specific classes of bioactive compounds. Thus, future studies will be performed in order to investigate the bioactive composition of apple by-products and their potential for obtaining valuable phytochemicals that can be used in the cosmetic, pharmaceutical and food industries.

## 4. Conclusions

Phenolic compounds and triterpene acids composition and bioactive properties (TP, TF and AC) vary considerably in the investigated apple cultivars. Caffeic, gallic, *p*-coumaric and ferulic acids were the dominant phenolic acids in Starkrimson, Jonathan, Florina, Montuan, Golden Delicious, Renet Simirenco, Spartan and Cretesc apple cultivars, while important amounts of (-)-epicatechin and quercetin were quantified in Starkrimson, Florina, Richard, Golden Delicious, Jonathan and Beliy Naliv apple cultivars, respectively. The highest values of AC and an implicitly high potential of polyphenols were observed for Starkrimson, Jonathan, Beliy Naliv and Richard cultivars, while Richard, Montuan, Golden Delicious, Idared and Beliy Naliv cultivars were characterized by the highest amounts of triterpene acids. PCA analysis discriminates between apple cultivars based on polyphenolic composition and allows the identification of specific polyphenolic markers for each cultivar. HCA analysis applied on triterpene acids composition allows clustering of the investigated apple cultivars depending on bioactive potential. This study represents one of the most comprehensive studies of the phenolic and triterpene compounds in Romanian and Moldavian apples and will help to further understand the polyphenolic and triterpene composition of apples and the roles of these compounds as health-promoting agents. This research will be extended in the future considering the investigation of a larger panel of phenolic compounds and triterpene acids in different cultivars from more regions and countries in order to identify the cultivars with the most technological interest.

## Figures and Tables

**Figure 1 foods-10-00267-f001:**
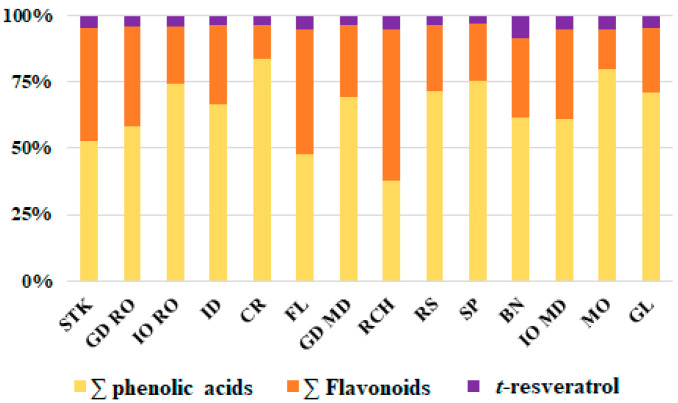
Ultra-high performance liquid chromatography (UHPLC) polyphenolic profile (phenolic acids, flavonoids and t-resveratrol) of different apple cultivars: STK-Starkrimson, GD RO-Golden Delicious from Romania, IO RO-Jonathan from Romania, ID-Idared, CR-Cretesc, FL-Florina, GD MD-Golden Delicious from Republic of Moldova, RCH-Richard, RS-Renet Simirenco, SP-Spartan, BN-Beliy Naliv, IO MD-Jonathan from Republic of Moldova, MO-Montuan, GL-Gloster.

**Figure 2 foods-10-00267-f002:**
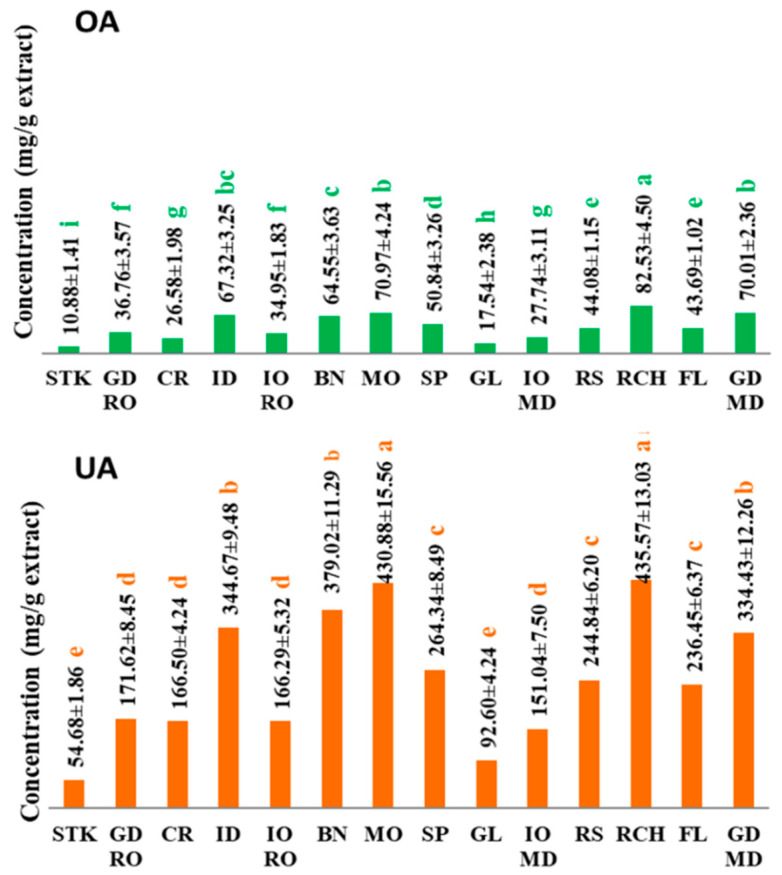
Oleanolic acid (OA) and ursolic acid (UA) content in extracts of different apple cultivars (STK-Starkrimson, GD-Golden Delicious from Romania, CR-Cretesc, ID-Idared, IO RO-Jonathan from Romania, BN-Beliy Naliv, MO-Montuan, SP-Spartan, GL-Gloster, IO MD-Jonathan from Republic of Moldova, RS-Renet Simirenco, RCH-Richard, FL-Florina, GD MD–Golden Delicious from the Republic of Moldova) as means ± SD. Different letters denote significant differences according to Duncan test *p* ≤ 0.05.

**Figure 3 foods-10-00267-f003:**
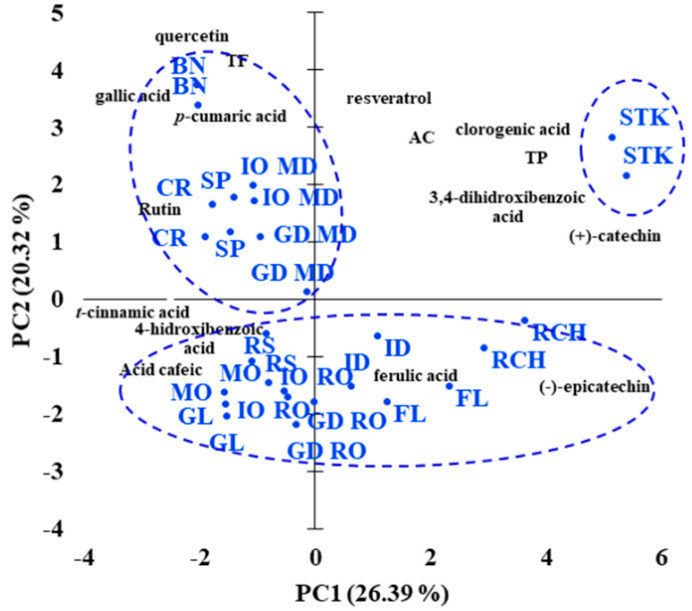
Bi-plot of the principal components PC1 and PC2, resulting from the PCA analysis of the polyphenolic composition of different apple cultivars.

**Figure 4 foods-10-00267-f004:**
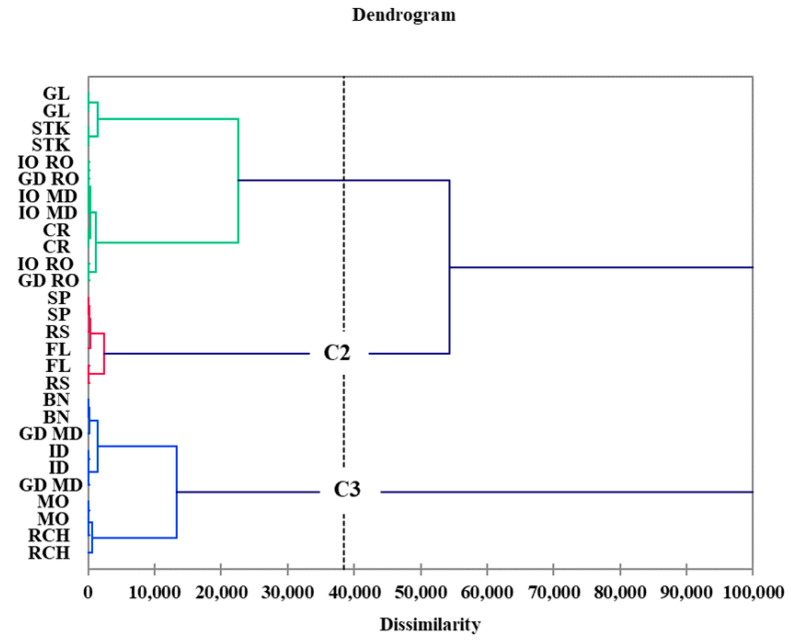
Dendrogram of the 12 apple cultivars based on triterpene composition.

**Table 1 foods-10-00267-t001:** Concentration (mg/100 g DW) of phenolic compounds in different apple cultivars.

Apple Variety	Phenolic Acids (mg/100 g DW)							
	Caffeic	Gallic	Ferulic	*p*-Coumaric	*p*-Hydroxybenzoic	3,4-Dihydroxybenzoic	*t*-Cinnamic	Chlorogenic
Beliy Naliv	9.90 ± 0.71 ^e^	13.44 ± 0.11 ^b^	2.28 ± 0.10 ^cd^	5.32 ± 0.11 ^bcde^	0.59 ± 0.06 ^a^	0.94 ± 0.08 ^a^	0.35 ± 0.07 ^a^	0.84 ± 0.11 ^c^
Cretesc	9.94 ± 0.23 ^e^	14.13 ± 0.27 ^b^	2.67 ± 0.16 ^cd^	16.31 ± 0.07 ^a^	0.43 ± 0.11 ^a^	0.84 ± 0.11 ^a^	0.35 ± 0.04 ^a^	1.02 ± 0.08 ^c^
Florina	13.57 ± 1.25 ^cde^	0.62 ± 0.18 ^h^	7.27 ± 0.94 ^a^	1.33 ± 0.94 ^e^	0.73 ± 0.13 ^a^	0.78 ± 0.13 ^a^	<LOQ ^b^	0.82 ± 0.21 ^c^
Gloster	9.43 ± 0.13 ^e^	7.73 ± 0.06 ^cd^	1.69 ± 0.06 ^de^	3.53 ± 0.04 ^de^	0.78 ± 0.07 ^a^	1.04 ± 0.06 ^a^	0.36 ± 0.06 ^a^	0.86 ± 0.06 ^c^
Golden Delicious MD	18.79 ± 0.73 ^bcd^	14.94 ± 0.09 ^b^	3.27 ± 0.29 ^bcd^	8.14 ± 1.14 ^bcd^	0.46 ± 0.01 ^a^	0.82 ± 0.10 ^a^	0.34 ± 0.01 ^a^	0.78 ± 0.18 ^c^
Golden Delicious RO	13.99 ± 3.3 ^cde^	2.84 ± 0.33 ^fg^	2.51 ± 0.61 ^cd^	4.59 ± 0.26 ^cde^	0.48 ± 0.07 ^a^	0.73 ± 0.11 ^a^	0.35 ± 0.01 ^a^	<LOQ ^d^
Idared	9.80 ± 0.85 ^e^	6.53 ± 1.06 ^de^	4.07 ± 0.30 ^bc^	4.21 ± 0.10 ^de^	0.60 ± 0.11 ^a^	1.22 ± 0.30 ^a^	0.35 ± 0.08 ^a^	2.30 ± 0.14 ^b^
Jonathan MD	10.41 ± 0.10 ^de^	13.14 ± 0.10 ^b^	0.39 ± 0.10 ^e^	5.42 ± 0.14 ^bcde^	0.84 ± 0.06 ^a^	0.86 ± 0.04 ^a^	0.33 ± 0.04 ^a^	1.06 ± 0.06 ^c^
Jonathan RO	24.19 ± 6.82 ^b^	2.24 ± 0.76 ^gh^	1.49 ± 0.50 ^de^	1.41 ± 0.06 ^e^	0.61 ± 0.15 ^a^	0.72 ± 0.03 ^a^	0.29 ± 0.07 ^a^	<LOQ ^d^
Montuan	47.23 ± 0.14 ^a^	3.04 ± 0.06 ^fg^	2.23 ± 0.04 ^cde^	1.09 ± 0.06 ^e^	0.60 ± 0.06 ^a^	0.84 ± 0.03 ^a^	<LOQ ^b^	<LOQ ^d^
Renet Simirenco	12.29 ± 0.46 ^cde^	9.80 ± 1.43 ^c^	4.87 ± 0.88 ^b^	5.43 ± 1.96 ^bcde^	0.48 ± 0.15 ^a^	0.62 ± 0.29 ^a^	0.33 ± 0.01 ^a^	0.48 ± 0.15 ^cd^
Richard	10.92 ± 1.38 ^cde^	0.49 ± 0.01 ^h^	2.02 ± 0.53 ^de^	2.22 ± 0.42 ^e^	0.38 ± 0.01 ^a^	1.09 ± 0.15 ^a^	<LOQ ^b^	0.44 ± 0.21 ^cd^
Spartan	9.83 ± 0.07 ^e^	21.74 ± 0.48 ^a^	1.82 ± 0.08 ^de^	9.45 ± 0.10 ^b^	0.46 ± 0.11 ^a^	0.72 ± 0.06 ^a^	<LOQ ^b^	0.96 ± 0.08 ^c^
Starkrimson	19.40 ± 1.18 ^bc^	4.50 ± 0.03 ^ef^	1.93 ± 0.69 ^de^	9.18 ± 3.58 ^bc^	0.53 ± 0.26 ^a^	1.38 ± 0.57 ^a^	<LOQ^b^	3.77 ± 0.53 ^a^

Data are expressed as means ± SD. Means followed by different lowercase letters in the column differ significantly according to the Duncan’s multiple range test at *p* ≤ 0.05; <LOQ: values bellow the quantification limit.

**Table 2 foods-10-00267-t002:** Concentration (mg/100 g dry weight (DW)) of flavonoids and *t*-resveratrol in different apple cultivars.

Apple Variety	Flavonoids				*t*-Resveratrol
	(+)-Catechin	(-)-Epicatechin	Quercetin	Rutin	
Beliy Naliv	0.96 ± 0.07 ^cd^	<LOQ ^f^	11.61 ± 0.10 ^a^	3.81 ± 0.07 ^a^	4.79 ± 0.06 ^a^
Cretesc	1.01 ± 0.04 ^cd^	<LOQ ^f^	3.67 ± 0.16 ^bcd^	2.15 ± 0.18 ^bc^	2.05 ± 0.10 ^c^
Florina	1.89 ± 0.32 ^bc^	19.65 ± 2.40 ^ab^	1.63 ± 0.92 ^de^	1.48 ± 0.18 ^cd^	2.69 ± 0.38 ^bc^
Gloster	0.64 ± 0.06 ^d^	4.69 ± 0.13 ^ef^	2.19 ± 0.10 ^cde^	1.04 ± 0.06 ^de^	1.70 ± 0.06 ^c^
Golden Delicious MD	0.90 ± 0.31 ^cd^	12.12 ± 3.63 ^cd^	5.45 ± 1.90 ^b^	0.38 ± 0.14 ^ef^	2.40 ± 0.12 ^bc^
Golden Delicious RO	1.24 ± 0.15 ^cd^	13.46 ± 1.25 ^bc^	1.28 ± 0.57 ^de^	0.48 ± 0.07 ^ef^	1.90 ± 0.13 ^c^
Idared	1.24 ± 0.21 ^cd^	6.90 ± 0.62 ^cdef^	5.03 ± 0.35 ^b^	<LOQ ^f^	1.25 ± 0.42 ^c^
Jonathan MD	1.00 ± 0.03 ^cd^	6.75 ± 0.07 ^cdef^	9.67 ± 0.04 ^a^	0.49 ± 0.08 ^ef^	2.76 ± 0.07 ^bc^
Jonathan RO	0.87 ± 0.18 ^cd^	6.75 ± 0.36 ^cdef^	1.10 ± 0.11 ^e^	0.27 ± 0.38 ^ef^	1.90 ± 0.13 ^c^
Montuan	0.83 ± 0.01 ^cd^	5.03 ± 0.07 ^def^	2.06 ± 0.06 ^cde^	2.50 ± 0.06 ^b^	3.42 ± 0.11 ^b^
Renet Simirenco	0.69 ± 0.17 ^d^	8.22 ± 0.83 ^cde^	1.28 ± 0.36 ^de^	1.87 ± 0.50 ^bcd^	1.60 ± 0.85 ^c^
Richard	2.88 ± 0.59 ^b^	22.36 ± 1.99 ^a^	1.54 ± 0.05 ^de^	<LOQ ^f^	2.38 ± 0.09 ^bc^
Spartan	1.04 ± 0.08 ^cd^	1.81 ± 0.06 ^ef^	9.91 ± 0.16 ^a^	<LOQ ^f^	1.72 ±0.07 ^c^
Starkrimson	4.08 ± 0.55 ^a^	25.03 ± 4.67 ^a^	4.17 ± 0.35 ^bc^	<LOQ ^f^	3.43 ± 0.60 ^b^

Data are expressed as means ± SD. Means followed by different lowercase letters in the column differ significantly according to the Duncan’s multiple range test at *p* ≤ 0.05; <LOQ: values bellow the quantification limit.

**Table 3 foods-10-00267-t003:** Correlation matrix between apples’ phenolic compounds profile and bioactive properties.

	∑ Phenolic Acids	∑ Flavonoids	TP	TF	AC
∑ phenolic acids	**1**				
∑ phenolic acids	−0.261	**1**			
TP	**0.547**	−0.028	**1**		
TF	−0.126	**0.776**	0.009	**1**	
AC	−0.137	**0.524**	0.457	**0.519**	**1**

Note: TP, total polyphenols; TF, total flavonoids; AC, antioxidant capacity (2,2-diphenyl-1-picrylhydrazyl (DPPH) assay). Bold values correspond to coefficients with values higher than 0.5

**Table 4 foods-10-00267-t004:** The main bioactive compounds identified in the studied apple cultivars and the associated biological and pharmacological activities which contribute to human health (according to the literature).

Main Bioactive Compounds	Main Associated Biological and Pharmacological Activities That Contribute to Human Health	References
caffeic acid	biological and pharmacological activities: antioxidant, anticancer, as part of a cancer treatment regime, treat certain viruses, including herpes and HIV	[38]
	marketed as a supplement to boost athletic performance and to aid in weight loss or as skin care serums	[41]
gallic acid	biological and pharmacological activities: antioxidant, antimicrobial, anti-inflammatory, anticancer, cardio-protective, gastro-protective, and neuroprotective effects, metabolic diseases	[35,42]
	marketed as flavouring agents and preservatives in the food industry	[43]
epicatechin	biological activities: myostatin inhibition, increase muscle growth and strength, improve blood pressure and reduce insulin resistance	[39,40]
	marketed as a supplement to enhance athletic performance	[44,45]
quercetin	effects on diabetes complications, Alzheimer’s, cardiovascular and liver diseases, Arthritis, Microbial Infections	[46]
	mediate anti-oxidative and anti-inflammatory activity and shows potential to decrease body fat percent and may improve exercise performance	[47]
ursolic acid	potential biological activities: reduce fat accumulation and increase muscle mass—obesity control, inhibits the proliferation of various cancer cell types, cardioprotective, anti-inflammatory, antibacterial, antidiabetic and neuroprotective activities. *NOTE*: there are not enough clinical studies to demonstrate the benefits to human health.	[36]
oleanolic acid	biological activities: hepato-protective, antitumor, antidiabetic, antimicrobial, antihypertensive and anti-inflammatory properties	[37]

## Data Availability

Not Available.

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
