# Peer review of "Profiling of Phenolic Compounds and Triterpene Acids of Twelve Apple (Malus domestica Borkh.) Cultivars"

_foods, 2021, doi:10.3390/foods10020267_

Round 1
Reviewer 1 Report
Dear Authors, the paper needs to be improved.
The novelty of the paper needs to be highlighted.
The EP needs to be improved.
The comparison with other similar papers needs to be increased. If few papers indicate different apple cultivars, the authors should put some comparison with other paper which study different fruit cultivars vs phenol composition (for example). Please see attachement.

Author Response
We want to thank to the reviewer for his valuable comments and observations, made directly on the text, which obviously contributed to improving the quality of our manuscript. The answers were made directly in the Review Report (Round 1), as reply. Also, we would like to thank for giving us a chance for publication of our paper, after the revision, considering the reviewers comments.

Reviewer 2 Report
The manuscript entitled “Profiling of phenolic compounds and triterpene acids in different apple (Malus domestica Borkh.) cultivars”, authored by Geana and colleagues, deals with the profiling of the bioactive compounds contained in different apple cultivars using both spectrophotometric and chromatographic methodogies. Moreover, authors employed statistical analysis (PCA and HCA) with the aim to discriminate the different cultivars from each other.
The main lack of the article is related to the methods used during the experimentations. The authors could have performed other simple spectrophotometric assays with the aim to evaluate the specific content of flavan-3-oils (DMAC assay) or to better measure the oxidant activity of the extracts (for example ABTS assay for the radical scavenging, FRAP and CUPRAC for the reducing activities, ORAC for the total antioxidant activity, etc..). However, I do not believe that the lack of these essays is a limit to the publication of the manuscript. Indeed, the manuscript investigates a really interesting topic, which can contribute to the commercial enhancement of specific local apple cultivars.
However, specific changes should be made. Below, a series of observations which I strongly suggest to follow:
- Please, check that Mallus domestica is written in italic all over the manuscript;
- keywords should be words not contained in the title of the manuscript. Their aim is to make the manuscript more visible when the universal research tools (such as PubMed) are employed. Some of the words provided by the authors are repetitive, as they appear both in the title and in the abstract. Please propose others and different keywords.
- Regarding section 2.4, please divide the method concerning HPLC analysis from those related to other assays. In particular, Folin Ciocolteau and AlCl3 methods should be placed in one, or two, different paragraphs. Moreover, authors should describe in detail the experimental conditions. Finally, DPPH assay is related to the measure of the radical scavenging property of fruits extracts (or in general their antioxidant activity). Consequently, a different subsection should contain the description of this method.
- The authors correctly evaluated the matrix effect and recovery during the extraction process. However, information regarding LOD (limit of detection) and LOQ (limit of quantification) is missing. These are important parameters for the validation of the HPLC method, and are especially essential if in the manuscript it is stated that a particular bioactive compound was not detected (for example, see Table 1). Authors can take useful information to calculate LOD and LOQ from this article, where these same parameters were measured using calibration curves (https://doi.org/10.3390/nu12040992). Briefly, LOD and LOQ may be calculated using the standard deviation of the response (Sy) and the slope (m) of the calibration curve [LOD = 3.3(Sy/m); LOQ = 10(Sy/m)]. m and Sy may be simply calculating using the MS Excel function “SLOPE” and “STEYX”. In order to perform the LOD and LOQ calculation, authors should use at least 15 different standard concentrations, with concentrations much lower than those detected for the samples.
- Table 1 should be placed on vertical page. Moreover, lowercase letters should be placed after the standard deviation of each value.
- Also in Table 2 the lowercase letters should be placed after the standard deviation of each value. Formatting errors are present in this table. For example, Starkrimson is written in bold (is there a reason for that?), a black line is present under its name, and a line is missing under resveratrol.
- In the caption of Figure 1, the meanings of acronyms should be given following the order in which they appear in the figure as histogram.
- The authors should divide the quantifications of oleanolic and ursolic acids into two different panels of the same figure. Moreover, the standard deviations appear to be missing. I am pretty sure and confident that the authors performed more than one replicate of their experiments. Consequently, standard deviations should be included in the graph. Remeber to add the information related to the two different panels in the caption of the new Figure.
- The authors reported only 35 bibliographic references, most of which very dated. Authors should include some more recent references.
Author Response
We want to thank to the reviewer for his valuable comments and observations which obviously contributed to improving the quality of our manuscript. Also, we would like to thank for giving us a chance for publication of our paper, after the revision, considering the reviewers comments. The answers are presented in the uploaded document.

Round 2
Reviewer 1 Report
Dear Authors,
the revision must be done carefully. I attach again a pdf file with some (only some) suggested changes, some of which have already been suggested, but not well motivated during the first review.
The sampling (number of samples, origin, month of harvesting) is fundamental for a work regarding PCA.
I suggest to the authors to take carefully into consideration the first and the second revision and the guide for authors for the abbreviations.

Author Response
We want to thank to the reviewer for his valuable comments and observations which obviously contributed to improving the quality of our manuscript.
Also, we would like to thank for giving us a chance for publication of our paper, after the second revision, considering the comments.
We have performed the answers directly on the text, as reply to the comments. We hope that all the comments were carefully considered. The changes in the manuscript are marked with in red.
Please see the attachment!
